

# Large-scale assessment of commensalistic–mutualistic associations between African birds and herbivorous mammals using internet photos

Peter Mikula[1], Jiří Hadrava[1,2], Tomáš Albrecht[1,3] and Piotr Tryjanowski[4]

[1] Department of Zoology, Faculty of Science, Charles University, Prague, Czech Republic
[2] Institute of Entomology, Biological Centre, Czech Academy of Sciences, České Budějovice, Czech Republic
[3] Institute of Vertebrate Biology, Czech Academy of Sciences, Brno, Czech Republic
[4] Institute of Zoology, Poznań University of Life Sciences, Poznan, Poland

Corresponding author
Peter Mikula,
petomikula158@gmail.com

## ABSTRACT

Birds sitting or feeding on live large African herbivorous mammals are a visible, yet quite neglected, type of commensalistic–mutualistic association. Here, we investigate general patterns in such relationships at large spatial and taxonomic scales. To obtain large-scale data, an extensive internet-based search for photos was carried out on Google Images. To characterize patterns of the structural organization of commensalistic–mutualistic associations between African birds and herbivorous mammals, we used a network analysis approach. We then employed phylogenetically-informed comparative analysis to explore whether features of bird visitation of mammals, i.e., their mean number, mass and species richness per mammal species, are shaped by a combination of host mammal (body mass and herd size) and environmental (habitat openness) characteristics. We found that the association web structure was only weakly nested for commensalistic as well as for mutualistic birds (oxpeckers *Buphagus* spp.) and African mammals. Moreover, except for oxpeckers, nestedness did not differ significantly from a null model indicating that birds do not prefer mammal species which are visited by a large number of bird species. In oxpeckers, however, a nested structure suggests a non-random assignment of birds to their mammal hosts. We also identified some new or rare associations between birds and mammals, but we failed to find several previously described associations. Furthermore, we found that mammal body mass positively influenced the number and mass of birds observed sitting on them in the full set of species (i.e., taking oxpeckers together with other bird species). We also found a positive correlation between mammal body mass and mass of non-oxpecker species as well as oxpeckers. Mammal herd size was associated with a higher mass of birds in the full set of species as well as in non-oxpecker species, and mammal species living in larger herds also attracted more bird species in the full set of species. Habitat openness influenced the mass of birds sitting on mammals as well as the number of species recorded sitting on mammals in the full set of species. In non-oxpecker species habitat openness was correlated with the bird number, mass and species richness. Our results provide evidence that patterns of bird–mammal associations can be linked to mammal and environmental characteristics and highlight the potential role of information technologies and new

media in further studies of ecology and evolution. However, further study is needed to get a proper insight into the biological and methodological processes underlying the observed patterns.

## INTRODUCTION

Commensalism (direct interactions between different organisms that are beneficial for one partner and neutral for the other) and mutualism (interactions that are mutually beneficial for both partners) are widespread forms of interspecific interactions between organisms (*Begon, Townsend & Harper, 2006*; *Goodale, Beauchamp & Ruxton, 2017*). The possibility of gaining some benefits through commensalistic and/or mutualistic associations may stimulate individuals of one species to actively seek, and associate with, individuals of another species. Among such interactions, one widespread but still understudied example of association is between birds and large terrestrial herbivorous mammals in Africa (*Dean & MacDonald, 1981*). Although living near large mammals might be sometimes costly since raptors have been recorded hunting alongside them (*Dean & MacDonald, 1981*), in general, interspecific associations seems to be advantageous (*Heymann & Hsia, 2015*). Many African birds use larger-bodied mammal hosts as perches and sometimes even as food sources, gleaning parasites and tissue from the host, improving the foraging efficiency of the birds, receiving more food and expending less energy than non-associated birds, or gaining increasing protection from predators (*Heatwole, 1965*; *Smith, 1971*; *Dean & MacDonald, 1981*; *Ruggiero & Eves, 1988*; *Koenig, 1997*; *Sazima et al., 2012*; *Ndlovu & Combrink, 2015*; *Goodale, Beauchamp & Ruxton, 2017*). The African herbivorous mammals are composed of many phylogenetic lineages with diverse life strategies, including their body masses and tendency to form herds (*Smith et al., 2004*; *Kingdon, 2015*), which makes them a moving system of islands and archipelagos across African ecosystems. The majority of previous studies investigating patterns in commensalistic–mutualistic interactions between African birds and large herbivores have focused only on single or a small number of species (*Hart et al., 1990*; *Koenig, 1997*; *Nunn et al., 2011*; *Ndlovu & Combrink, 2015*; *Kioko et al., 2016*) and no wide-scale study of patterns in bird–mammal interactions has been done on African fauna. Hence, a large-scale and multitaxonomical approach is useful when investigating patterns in bird–mammal interactions to avoid problems with interpretation and generalization of relationships which may be area- or taxa-specific.

Many types of heterospecific relationships, including both commensalism and mutualism, are depicted as complex webs comprising several interacting species, rather than as isolated interactions between species pairs (*Bascompte et al., 2003*). As a result, the structure of such community networks exhibit a specific arrangement of interactions rather than random inter-specific interactions. This specific type of community organization is called "nestedness", and is characterized by (1) more selective species (i.e., those that

have few interactions) that tend to interact only with subsets of those species interacting with more generalized species (i.e., that have many interactions), (2) generalized species that tend to interact with other generalized species, forming a highly cohesive core of interacting species, and (3) the absence of specialized species that interact only with other specialized species (*Bascompte et al., 2003*; *Guimarães et al., 2006*; *Joppa et al., 2010*; *Sazima et al., 2012*). Although some studies suggested that different measures of nestedness should be used and results should be compared with a null model, which is able to eliminate the effect of abundance inequality (*Vázquez et al., 2007*; *Ulrich, Almeida-Neto & Gotelli, 2009*), the nested web structure is still common, mainly in mutualistic but also commensalistic networks (*Lewinsohn et al., 2006*; *Joppa et al., 2010*; *Sazima et al., 2012*; *Sáyago et al., 2013*). However, studies involving birds as interactors have had mixed results. For instance, while a highly nested structure was found for cleaning associations between birds and their mammal hosts in Neotropical regions (*Sazima et al., 2012*), the hummingbird–plant association web was not nested (*Vizentin-Bugoni, Maruyama & Sazima, 2014*).

Although some bird species may be involved in mutualistic interactions with mammal herbivores, the majority of African birds rather form commensalistic sitting associations with mammals (*Dean & MacDonald, 1981*; *Kioko et al., 2016*). The number and diversity of birds directly interacting with (i.e., sitting on) mammals could be influenced by extrinsic factors, such as host body mass, mainly due to limited space available for birds and the load-carrying capability of mammal species. Moreover, larger mammals or mammals living in larger herds could be more visible to birds and/or disturb more insects and other small animals, and subsequently may attract a wider and more abundant community of birds looking for food (*Mooring & Mundy, 1996*; *Nunn et al., 2011*; *Kioko et al., 2016*). Environmental factors such as vegetation structure have also been shown to have an effect on species richness, distribution, and abundance of birds and mammals (*Terborgh, 1977*; *James & Wamer, 1982*; *Rahbek, 1995*; *Tews et al., 2004*; *Jankowski et al., 2013*), potentially shaping associations between birds and herbivorous mammals (*Hart et al., 1990*; *Kioko et al., 2016*). For instance, birds, such as waterbirds, living near water sources are expected to interact more often with mammals inhabiting these habitats, such as common hippopotamus *Hippopotamus amphibius* (*Dean & MacDonald, 1981*). Moreover, it seems that in Africa, some associations are more common in open areas such as savannah and grassland than in forests (*Dean & MacDonald, 1981*).

The only examples of African birds exhibiting obligate mutualistic associations with mammals are the small-bodied passerines, oxpeckers (Buphagidae), being two extant species, yellow-billed oxpecker *Buphagus africanus* and red-billed oxpecker *B. erythrorhynchus*. Here, the species association features may differ from other birds since the feeding ecology of oxpeckers and their presence on host species has been found to be strongly correlated with the character of host infestation by ectoparasites (*Hart et al., 1990*; *Nunn et al., 2011*). Mammal species and individuals differ significantly in tick infestation (*Gallivan & Horak, 1997*); the distribution of oxpeckers can thus potentially follow tick density rather than mammal body mass and herd size *per se*. For instance, oxpeckers may prefer to visit mammal species or individuals inhabiting woody and scrubby areas with reportedly higher tick density than open grassland areas (*Semtner & Hair, 1973*; *Carroll &*

*Schmidtmann, 1996*), larger mammals supporting a higher tick abundance than smaller ones (*Koenig, 1997*; *Nunn et al., 2011*) or, smaller mammals with a higher tick number to body mass ratio (*Hart et al., 1990*). In addition, a preference for mammals living in larger herds could be an adaptive strategy for oxpeckers since the decreasing distance between host individuals makes feeding more efficient (*Mooring & Mundy, 1996*).

To investigate large-scale patterns of bird–mammal associations, extensive data collection from free online sources may be useful. During the last decade, the engagement of volunteers in scientific projects, so-called citizen science, has became an integral part of current ecological and evolutionary research (*Bonney et al., 2009*; *Silvertown, 2009*; *Dickinson, Zuckerberg & Bonter, 2010*). Approaches range from the collection of internet data uploaded by the public to active participation and collaboration with scientists (e.g., via online platforms) on a wide range of projects. Rapid technological development and the expanding access of the public to both internet and recording devices, such as cameras or smartphones, around the world have increased the accessibility, immediacy and extent of data sharing. Online data collected by the public can represent a useful resource for expansion of scientific knowledge on rare or poorly studied phenomena (e.g., *Mikula et al., 2016*) and facilitate cost-effective and rapid large-scale data mining which may supplement or even challenge conventional practices in science (*Leighton et al., 2016*; *Dylewski et al., 2017*). Despite the increasing number of such studies, material uploaded on the internet by the public is still an underexploited data source for studies in ecology and evolution.

Here, we used photos collected using the web-based search engine Google Images to investigate some aspects of commensalistic–mutualistic associations between African birds and herbivorous mammals. In contrast to the majority of previous field studies that focused only on spatially and taxonomically restricted systems (e.g., *Ndlovu & Combrink, 2015*; *Kioko et al., 2016*) or did not quantify these relationships (*Dean & MacDonald, 1981*), we provide, to the best of our knowledge, the first comprehensive investigation of patterns in associations between African birds and mammals at large spatial and taxonomic scales. Firstly, the structure of the association web between African birds and mammals was visualized and analysed to investigate frequencies of association between particular bird and mammal species and whether bird–mammal interactions were arranged in a nested pattern. Then, we employed phylogenetically-informed comparative analysis to explore whether patterns in bird visitation of mammals (i.e., mean number of birds and mean total mass of birds per mammal individual, and total number of bird species per mammal species) are linked to a combination of mammal host body mass and herd size, and habitat openness. Because it is often difficult to discriminate between mutualistic and commensalistic associations even in the field (*Goodale, Beauchamp & Ruxton, 2017*), we present analysis done on three sets of species: (1) all species, (2) non-oxpecker bird species, having apparently mainly commensalistic relationships with mammals, and (3) oxpeckers, the only obligate mutualists with mammals in Africa.

## MATERIALS AND METHODS

### Data searching

To collect a large dataset of spatially and taxonomically distributed data on bird–mammal associations, we did an extensive internet search for photos on Google Images. *Jarić et al. (2016)* pointed out that internet searching based on English common names produced more results than when scientific names were used. Moreover, since results of searches using English and scientific names were highly correlated, we decided to search only for English names of birds and mammals, although this could restrict the geographical coverage and decrease the use of records from non-English-speaking countries. To capture relative frequency of each bird–mammal association, our search phrase typically contained the name of the bird species combined with the name of the species/genus of larger-bodied African mammal herbivore (>10 kg). Our list of taxon names was based mainly on association reports reviewed by *Dean & MacDonald (1981)*, but we also included searches for associations not reported by them; for the complete index (see Appendix S1). If no bird visitors were recorded for some mammal species, we repeated the search using a more general "bird" or "birds" term to find whether, at least in some cases, these mammals were visited by birds. We also used this searching phrase for species where few interactions had already been found by a word combination search. However, we used only results revealing new, typically rare, bird–mammal associations, hence avoiding significant bias in the search in favour of common or well-recognized associations.

The Google searches for photos for each combination of bird and mammal taxa were conducted separately, and for each combination we aimed to collect as many photos as possible until the search produced only a small proportion of photos with relevant content. For common species it is virtually impossible to collect all available photos, so this solution represented a trade-off between the number of available relevant photos and the time spent searching for new photos; however, we consider that the proportion of available photos sampled was similar in all species. This procedure standardized our data for analysis, increasing possibility that variations in the frequency of internet photos in our dataset may reflect proportional differences in real animal abundance and/or extent of spatial distribution. However, the potential influence of the "charisma" of a species on its appearance on the internet requires cautious interpretation. We only analyzed photos in which birds were in direct contact with the bodies of a mammal, excluding cases where the birds were only feeding or flying near the mammal. We did not include photos of mammals without birds in our data set, even if such individuals were visible in photos.

We focused only on free-living, non-domesticated mammal species in sub-Saharan Africa. We also excluded photos where birds were observed on captive African mammals outside Africa (e.g., zoos). When photos were part of a series, we chose the one showing the highest number of associated birds/mammals. Paintings and photos which were suspected to be photomontages were ignored (<1% of all photos). To limit other sources of bias, photos suspected to be shared by multiple sources were briefly checked to see whether they had already been included (all photos were collected exclusively by one author, PM,

enabling us to do this consistently). We were particularly careful when working with unusual photos that people might prefer to share, e.g., a mammal individual covered by a large number of birds or "cute" or interesting animal species and scenes. However, it is still possible that a small number of duplicates remained undetected because we did not cross-check all possible combinations of photos. On the other hand, such cases were quite rare, and it seems that, for the volume of photos we collected, online sharing of photos would not substantially bias results obtained from a Google search compared to field data. Similarly, *Leighton et al. (2016)* found that the proportion of black colour morphs in black bear subspecies collected from a Google search was highly correlated with that from fieldwork, suggesting that online photos do not substantially over-represent bears with atypical colouration in particular subspecies.

Although there could be an inherent bias toward photos with larger numbers of birds on mammals caused by a preference for photographers to publish such photos, this should apply to all species and not affect relative differences. Even if only some photographers did this, we again do not expect any consistent bias in our data because photos originated from numerous authors. Furthermore, the probability of photographing a bird on a mammal may be related to the habitat structure (e.g., dense versus open habitats) and may underestimate involvement of species living in more closed habitats. Because a substantial proportion of the photos came from amateur photographers we expected less of a bias toward rare species or other unusual occurrences, as compared to professional photographers or researchers (see also *Dylewski et al., 2017*).

To investigate the spatial patterns in our dataset, we included only photos with the location given to at least country level. For each record, we specified the geographical location as accurately as possible. When the location was only at the country level, coordinates were taken as the centre of the mammal and/or bird species distribution in that country (accessed on http://www.iucnredlist.org and http://www.hbw.com).

## Bird and mammal characteristics

Each individual host mammal with birds observed on it represents a basic unit which was assessed separately. In the case of photos showing several mammals, each with one or several birds, each mammal individual was scored as a separate case (such cases represented <15% of all cases). Altogether we collected information on three response variables and three predictor variables:

(a) Response variables

*Mean number of birds per mammal individual* (hereafter referred to as "number of birds"): We counted the number of individual birds sitting on each individual mammal. For each mammal species, we then calculated the mean number of birds sitting on them.

*Mean total mass of birds per mammal individual* (hereafter referred to as "mass of birds"): The mean body mass of each bird species associated with individual mammals was obtained from the online edition of "Handbook of the Birds of the World" (*HBW Alive, 2016*). We used mean body mass for nominate subspecies and did not distinguish between sexes. For each mammal species, we then calculated the mean mass of birds sitting on them.

*Total number of bird species per mammal species* (hereafter referred to as "number of bird species"): We also collected information on the total number of bird species on each mammal species across the entire pool of photos in order to assess overall bird species richness hosted by them.

(b) Predictor variables

*Mammal body mass*: Information on body mass (in kilograms) of each mammal species was taken from the online "Encyclopedia of Life" (*EOL, 2016*). We used mean body mass for nominate subspecies and did not distinguish between sexes.

*Mammal herd size*: Herd size was estimated as the number of visible mammal individuals (conspecific or heterospecific) in each analysed photo. For each mammal species, we then calculated mean herd size.

*Habitat openness*: We defined the surrounding habitat for each photo and subjectively scored it into four distinct classes of openness, from 1 for most open habitats to 4 for very closed habitats: (1) near water (e.g., swamps, lakes and other water sources with typically very open vegetation cover), (2) open habitats (e.g., grassland, semi-deserts, open savannahs), (3) higher mosaic vegetation cover (e.g., woodland savannah, shrubland and bush) and (4) higher dense vegetation cover (e.g., woodland and forest). Then, we calculated mean values of habitat openness for each mammal species. We did not assess habitat from highly magnified (zoomed) photos because the identification would not be reliable.

## Photo zoom

Because we used several variables which are expected to be strongly influenced by photo zoom, all analyzed photos were scored according to their zoom on a three-point scale: (1) very zoomed photos (only part of mammal body was visible, e.g., head or hind legs with part of the belly), (2) medium zoomed photos (complete or almost complete mammal body was visible and free space on the photo constituted less than one mammal body length on each side), and (3) unzoomed photos (complete mammal body was visible and free space of more than one body length on each side was present).

## Statistical analysis
### Analysis of association web structure

In bird–mammal association web analyses, we included only associations where both mammals and birds were identified to species level. To avoid pseudoreplication, our basic unit for analysis was the number of cases and not the number of individuals, i.e., if several birds were observed on one individual mammal we considered this as one case. Because all associations were sampled proportionally equally, we minimized biases in measures of specificity resulting mainly from variation in species abundance and extent of the spatial distribution.

The structure of the bird–mammal association network for each species was visualized by the "plotweb" function and the network was analysed using the R-package *bipartite* (*Dormann et al., 2009*). To test whether a nested structure exists for our set of bird–mammal associations and whether there are differences in web structure between mutualists and commensals, we calculated nestedness of the bird–mammal association network for three

sets of species: (1) the full set of species, (2) non-oxpecker birds (mainly commensals), and (3) oxpeckers (mutuals). We used quantitative NODF (Nestedness metric based on Overlap and Decreasing Fill) (*Almeida-Neto & Ulrich, 2011*). Values of NODF range from zero in a non-nested web to 100 in a perfectly nested web (*Almeida-Neto et al., 2008*). We tested whether our network was significantly nested by comparing our network weighted NODF with a weighted NODF of 1000 networks generated randomly using a null model "swap.web" (*Dormann et al., 2009*).

### Effect of photo zoom

Because photo zoom may have had a significant effect on several included variables, we looked at the potential effect of photo zoom on variable estimates. First, greater zoom levels could bias estimation of mammal herd size because some individuals could be omitted from the photo. Our analyses revealed that species-specific herd size estimated from only unzoomed photos closely followed those from all photos (Pearson correlation $r = 0.85$, $p = 0.002$, $N = 10$ species; only species with a minimum of three unzoomed photos taken, data log-transformed before analysis). Second, photos depicting only part of the mammal body might underestimate the actual number of birds present on the mammal body because other individual birds can occupy unshown parts of the body at the same time. When several photos of the same scene were available, we tried to avoid this issue by using the photo with the maximum number of interacting birds and mammals. More importantly, all or almost all of the mammal body was visible in ~65% of all our records, and we found that the number and mass of birds estimated from very zoomed photos was strongly correlated with overall estimates (Pearson correlation $r = 0.61$, $p = 0.012$ and $r = 0.72$, $p = 0.002$, respectively, $N = 16$ species; species with minimum of three estimates, data log-transformed before analyses).

### Regression analysis

Although both generalist and specialized species can feed from the surface of the mammals (*Dean & MacDonald, 1981*; *Sazima et al., 2012*; *Ndlovu & Combrink, 2015*), due to the strong association of oxpeckers with this association, we decided to make analyses for the three bird groups previously mentioned (1) all species, (2) non-oxpecker bird species, and (3) oxpeckers. We used a species-based approach with the same-species individuals being used to calculate species means. Because analysis was controlled for phylogenetic relationships in mammals, we accepted only records identified to species level. However, to minimize losses of information, we also accepted observations for birds which were indistinguishable between two possible species; in these cases, body mass was calculated as the mean body mass of both species.

We used phylogenetic generalized least-squares (PGLS) regressions using Pagel's lambda transformation of a correlation structure to estimate the effects of mammal and environmental characteristics on bird-associated characteristics after controlling for phylogenetic relatedness of the mammal species (*Paradis, 2011*). Animal characteristics, such as body mass and behavioural patterns including social and feeding behaviour, have been found to be influenced by shared ancestry (*Smith et al., 2004*; *Kappeler et al., 2013*; *Lefebvre, Ducatez & Audet, 2016*); identified patterns may be determined by a

phylogenetically non-random set of species since phylogenetically related taxa thus have a higher probability of sharing characteristics from a common ancestor than do distant ones. The PGLS approach represents an extension of GLMs, accounting for the statistical non-independence of data points as a result of common ancestry of species (*Pagel, 1999*; *Freckleton, Harvey & Pagel, 2002*) and allows the estimation (via maximum likelihood) of the phylogenetic scaling parameter lambda ($\lambda$). A high value of lambda (i.e., $\lambda = 1$) indicates that species' traits covary in direct proportion to their shared evolutionary history, whereas $\lambda = 0$ indicate no phylogenetic relatedness (*Freckleton, Harvey & Pagel, 2002*). The maximum likelihood estimate of $\lambda$ thus provides a measure of the importance of phylogenetic relationships on the association between studied variables. Because sample sizes varied significantly among species, we weighted all analysis by sample size to adjust for potential effects of unequal sampling effort on estimates of true species means (*Garamszegi & Møller, 2010*).

We built multi-predictor models where the response variables were bird characteristics (number of birds, mass of birds, and number of bird species) and predictors were mammal (body mass and herd size) and environmental (habitat openness) characteristics. However, because species richness measures are typically influenced by the sample size effect (*Gotelli & Colwell, 2001*), we checked for correlation using Spearman correlation coefficient. We found a strong correlation between the number of bird species recorded on a mammal species and the number of photos available for this species ($r_s = 0.85$). For the subset of oxpeckers we used only mass of oxpeckers as a response variable because this family consists of only two extant species, which often live in allopatry (hence, causing low variability in species richness per mammal species) and are of similar body mass (causing the mean number of oxpeckers, even if both species were recorded on the same mammal species, to be highly correlated with their mean mass).

Both predictor and response variables were log-transformed prior to analyses. For a few species of mammals, some data, mainly on habitat openness, were missing; to avoid loss of such species from analyses we replaced missing values by mammal family averages. As suggested by *Forstmeier & Schielzeth (2011)* we present the models using all species because they clearly show the range of predictors included plus a balanced representation of non-significant results.

Reconstruction of the phylogenetic tree of African mammals was based on recent extensive data published by *Hedges et al. (2015)* (available online at http://www. biodiversitycenter.org/ttol). Normality of regression residuals after fitting the full models was checked using the Shapiro–Wilk test, revealing no violation of the assumptions of normality. The only exception was the model for oxpeckers where the use of raw untransformed variables resulted in a normal distribution of model residuals. PGLS regressions were performed using the *nlme* and *ape* packages (*Pinheiro et al., 2014*; *Paradis, Claude & Strimmer, 2015*). All data were statistically analysed in R v. 3.3.3 (*R Development Core Team, 2017*).
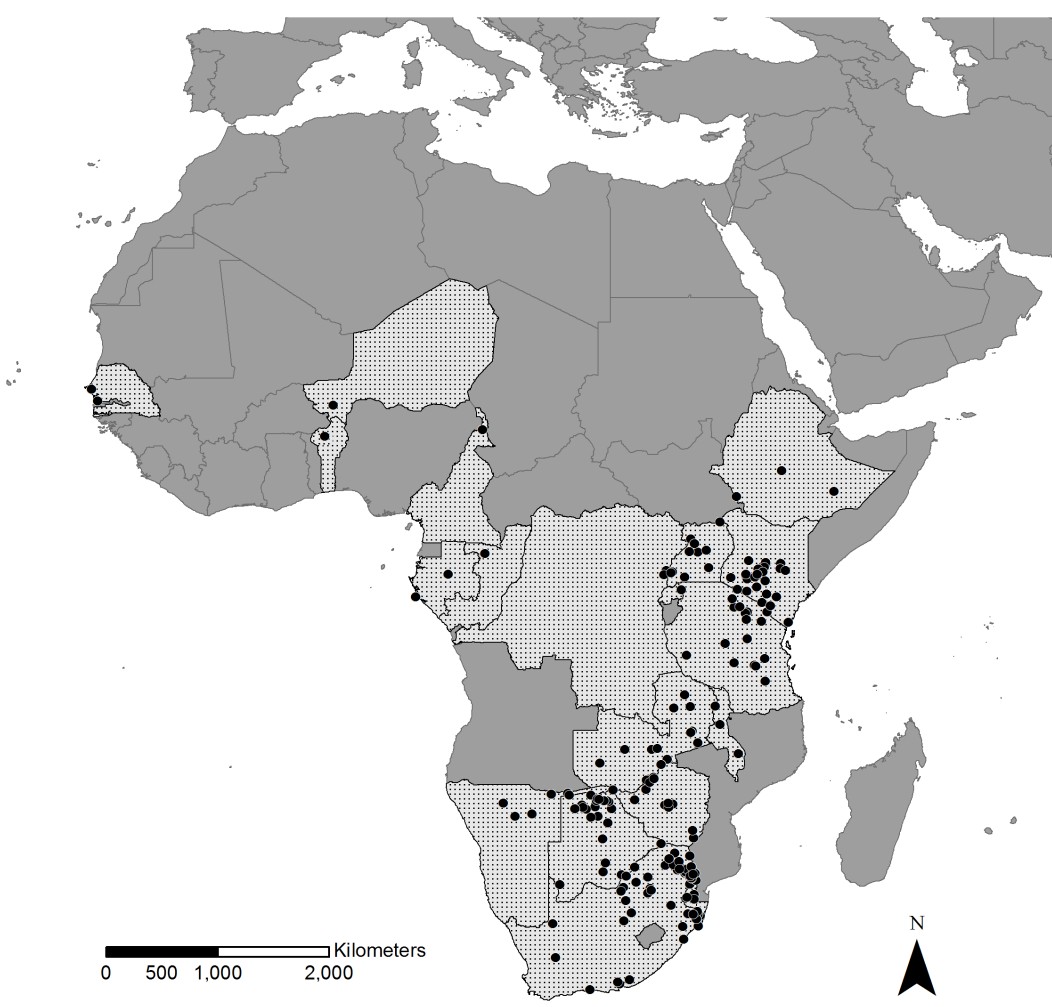

**Figure 1** **Geographical distribution of recorded bird–mammal associations.** Stippled areas are countries for which we had bird–mammal data and dots represent locations where photos were taken. Most records are distributed in East and Southern Africa whereas only restricted numbers of records originate from Central and West Africa.

# RESULTS

## Taxonomic diversity and spatial distribution of bird–mammal associations

In total, we collected information on 2,169 bird–mammal associations of 4,840 individual birds, belonging to at least 48 bird species of 21 families, with 31 species of wild-living African mammals of seven families. This dataset contains records from regions across sub-Saharan Africa with the majority of records from the open and relatively well studied areas of East and Southern Africa (Fig. 1). Only a small number of records came from West and Central Africa (<2%).

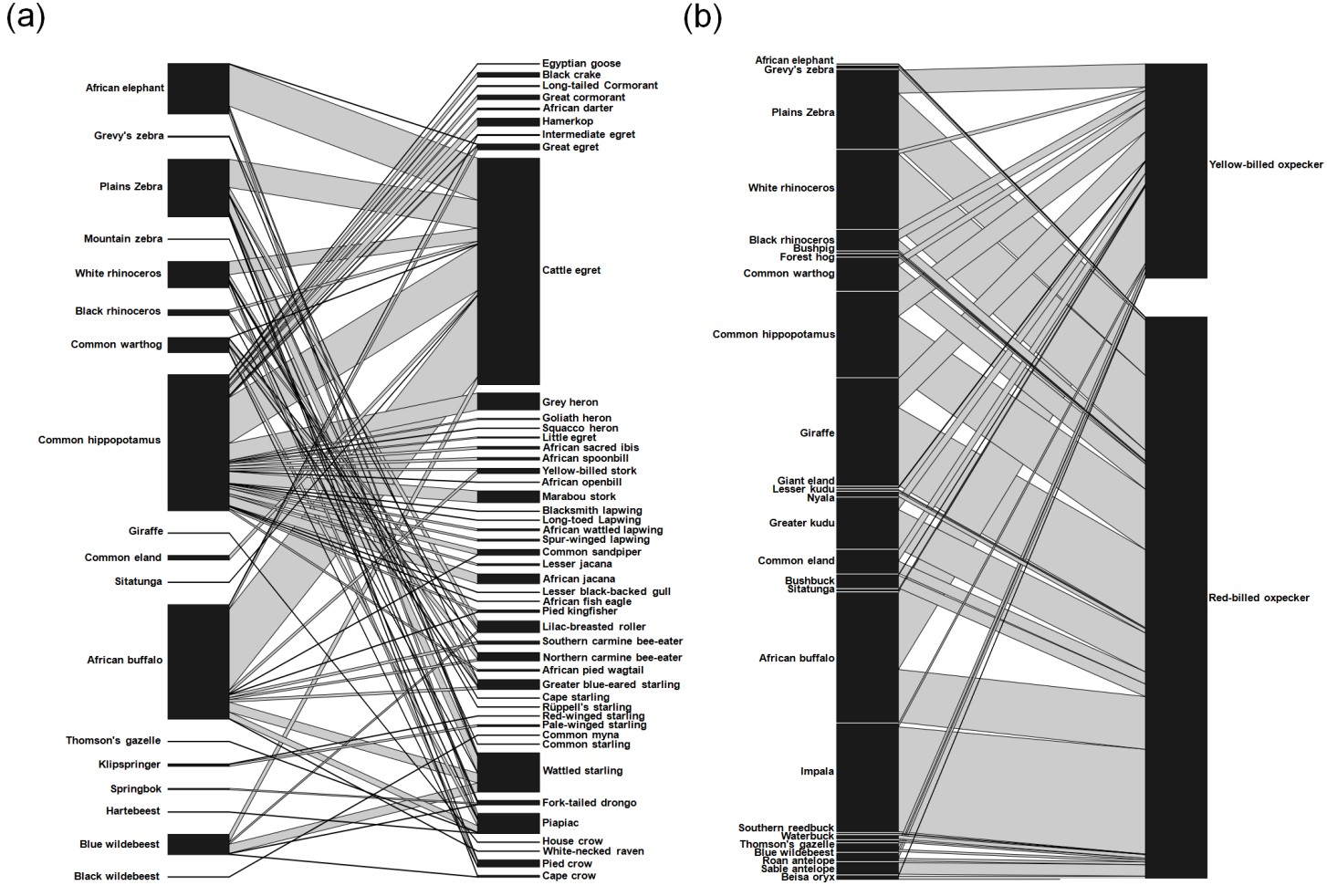

**Figure 2** **Quantitative bird–mammal association webs for (A) non-oxpecker species and (B) oxpeckers only.** For each web, the left bars represent the frequency with which each mammal species is visited by birds, and right bars represent the number of associations for each bird species. Associations for all mammal and bird species are ordered according to phylogenetic relationships.

## Bird–mammal association web

We included 2,147 associations in web analyses where both bird and mammal interactors were identified to species level (data in Appendix S2). These associations represent 123 different association types (i.e., combinations of different bird and mammal species). Of these, 66 (53.7%) association types were not reported by *Dean & MacDonald (1981)* and some may represent new associations, previously not reported in literature (see Appendix S2).

Of all cases, 672 cases (31.3%) included birds other than oxpeckers, detected on 18 species of mammals (Fig. 2A). These included cattle egret *Bubulcus ibis* (51.5% of non-oxpecker cases), wattled starling *Creatophora cinerea* (8.9%) and piapiac *Ptilostomus afer* (4.6%). In all records, the most visited mammals were common hippopotamus (31.0%), followed by plains zebra *Equus quagga* (13.1%) and African elephant *Loxodonta africana* (11.5%) (Fig. 2A).

We also found 1,475 cases (68.7%) that included oxpeckers: yellow-billed oxpecker (407 cases, 27.5% of oxpecker records) was observed on 16 mammal species whereas red-billed oxpecker (1,068 cases, 72.4%) was observed on 24 species of mammals (Fig. 2B). Yellow-billed oxpecker was most often associated with African buffalo *Syncerus cafer* (35.9% of all species-specific cases), giraffe *Giraffa camelopardalis* (13.5%) and hippopotamus (11.3%). The mammal species most often visited by red-billed oxpecker was impala *Aepyceros melampus* (18.6%), followed by giraffe (13.9%) and white rhinoceros *Ceratotherium simum* (13.3%) (Fig. 2B).

When all species were analysed together, we found that the association web between birds and their mammal hosts had rather low level of nestedness (NODF = 24.66) and did not differ significantly from values expected under the null model ($p = 0.77$). When separate analyses for oxpeckers and for the remaining species were carried out, we found that web nestedness was higher in oxpeckers than in the other species and differed significantly only for oxpeckers (NODF = 32.55, $p = 0.017$; other species NODF = 10.63, $p = 0.999$).

## Relationship between bird, mammal and environmental characteristics

In the full set of species ($N = 31$ species of mammals), the PGLS model analysing relationships between the number of birds and mammal and environmental characteristics revealed a positive correlation only with mammal body mass (model log likelihood = $-3.977$, $\lambda = 0.663$) (for complete results see Table 1; data in Appendix S3). The mass of birds was positively correlated with mammal body mass and herd size, and a higher mass of birds was also associated with more open areas (full model statistics: log likelihood = 5.653, $\lambda = -0.967$). The number of bird species was positively correlated with mammal herd size, and more species were recorded in open areas (log likelihood = $-42.769$, $\lambda = 0.501$).

In non-oxpecker bird species ($N = 19$ species of mammals), we found that the number of birds was higher in closed habitats (log likelihood = $-1.406$, $\lambda = 0.974$). The mass of birds was positively correlated with mammal body mass and herd size, and a higher mass of birds was also associated with more open areas (log likelihood = 2.920, $\lambda = -0.539$). A higher number of bird species was also associated with more open areas (log likelihood = $-22.630$, $\lambda = 0.634$). In oxpeckers ($N = 26$ species of mammals), we found a significant relationship between the mass of oxpeckers and mammal body mass (log likelihood = $-123.685$, $\lambda = -0.852$).

## DISCUSSION

Africa probably harbours the world's richest commensalistic–mutualistic associations between birds and larger-bodied mammal herbivores (*Dean & MacDonald, 1981*; *Ruggiero & Eves, 1988*), enabling us to look at general patterns in such associations. On the large set of photos collected from Google Images, we found that commensalistic–mutualistic associations between African birds and mammals are quite complex, involving many interacting species of both birds and mammals. Despite this, the web structure for African birds and mammals derived from internet photos was only weakly nested, even for oxpeckers. Furthermore, phylogenetically-informed comparative analysis revealed that

**Table 1 Relationships between bird, mammal and environmental characteristics.** Relationships between number of birds and mass of birds per mammal individual, and number of bird species per mammal species and mammal (body mass and herd size) and environmental characteristics (habitat openness), respectively, for the full set of species, non-oxpecker species, and oxpeckers, after correcting for phylogenetic relationships of the mammal species using phylogenetic generalized least square regression (PGLS). Statistically significant relationships are highlighted in bold.

| Model | Slope | SE | *t*-value | *p*-value |
|---|---|---|---|---|
| **All species** | | | | |
| *Number of birds* | | | | |
| Intercept | 0.185 | 0.239 | 0.773 | 0.446 |
| Mammal body mass | 0.094 | 0.025 | 3.740 | <**0.001** |
| Mammal herd size | −0.057 | 0.048 | −1.190 | 0.245 |
| Habitat openness | 0.139 | 0.141 | 0.983 | 0.334 |
| *Mass of birds* | | | | |
| Intercept | 4.148 | 0.147 | 28.248 | <0.001 |
| Mammal body mass | 0.242 | 0.019 | 12.912 | <**0.001** |
| Mammal herd size | 0.441 | 0.050 | 8.882 | <**0.001** |
| Habitat openness | −0.973 | 0.139 | −7.009 | <**0.001** |
| *Number of bird species* | | | | |
| Intercept | 1.868 | 0.834 | 2.239 | 0.034 |
| Mammal body mass | 0.128 | 0.087 | 1.469 | 0.153 |
| Mammal herd size | 0.474 | 0.169 | 2.807 | **0.009** |
| Habitat openness | −1.423 | 0.487 | −2.921 | **0.007** |
| **Non-oxpecker species** | | | | |
| *Number of birds* | | | | |
| Intercept | 0.573 | 0.328 | 1.747 | 0.101 |
| Mammal body mass | −0.039 | 0.042 | −0.937 | 0.364 |
| Mammal herd size | 0.060 | 0.031 | 1.964 | 0.068 |
| Habitat openness | 0.0432 | 0.147 | 2.949 | **0.010** |
| *Mass of birds* | | | | |
| Intercept | 5.791 | 0.245 | 23.602 | <0.001 |
| Mammal body mass | 0.170 | 0.028 | 6.066 | <**0.001** |
| Mammal herd size | 0.201 | 0.048 | 4.228 | <**0.001** |
| Habitat openness | −1.752 | 0.130 | −13.533 | <**0.001** |
| *Number of bird species* | | | | |
| Intercept | 4.368 | 0.937 | 4.660 | <0.001 |
| Mammal body mass | −0.157 | 0.118 | −1.329 | 0.204 |
| Mammal herd size | 0.203 | 0.115 | 1.769 | 0.097 |
| Habitat openness | −2.206 | 0.363 | −6.082 | <**0.001** |
| **Oxpeckers** | | | | |
| *Mass of birds* | | | | |
| Intercept | 80.886 | 56.663 | 1.427 | 0.168 |
| Mammal body mass | 0.019 | 0.005 | 3.697 | **0.001** |
| Mammal herd size | 15.429 | 12.546 | 1.230 | 0.232 |
| Habitat openness | 11.002 | 20.084 | 0.548 | 0.589 |

mammal body mass, herd size and habitat openness are important predictors of some patterns in bird–mammal associations, although with a different relationship for particular bird groups and characteristics. Below, we discuss the most significant results and we propose some biological explanations for the detected patterns. However, we are aware of several limitations of the approach used and, hence, interpretations and extrapolations of our results to real biological systems must be cautious. Our arguments should be understood in the context of the method we used.

## Structure of bird–mammal association web

Visualization of the bird–mammal association web revealed that the majority of bird species were associated with larger-bodied herbivores, such as common hippopotamus, African buffalo, plains zebra and African elephant, that occupy mainly open habitats (see mammal body and habitat openness scores in Appendix S3). We found that ∼60% of bird species in the data set were associated with water or were near water ecosystems with many of them recorded sitting exclusively on common hippopotamus. The most common bird interactor was cattle egret which is renowned for its widespread cooperative behaviour with African mammals (*Dean & MacDonald, 1981*; *Ruggiero & Eves, 1988*; *Kioko et al., 2016*; *Goodale, Beauchamp & Ruxton, 2017*). It is possible that the higher diversity of birds on larger-bodied mammals may be linked to the higher load-carry capacity of these mammals (hence, they can carry both smaller and larger bird species), and they can also provide more feeding opportunities, e.g., by flushing more prey (*Wahungu, Mumia & Manoa, 2003*; *Kioko et al., 2016*). Obviously, habitat type where particular bird and mammal species co-occur also has an important effect on bird–mammal association webs, facilitating some associations while limiting others (*Heymann & Hsia, 2015*; *Kioko et al., 2016*).

We found that obligate mammal mutuals, oxpeckers, visited a higher number of mammal species compared with the rather casual mammal-associated bird mutuals in the Neotropical region (*Sazima et al., 2012*). In agreement with previous field studies, the analysis of internet photos showed that oxpeckers were very often associated with larger-bodied mammals (*Mooring & Mundy, 1996*; *Koenig, 1997*; *Nunn et al., 2011*; *Ndlovu & Combrink, 2015*). Yellow-billed oxpecker was most often associated with African buffalo, giraffe and hippopotamus, while red-billed oxpecker was most often associated with impala, giraffe and white rhinoceros. With the exception of impala, the species mentioned are among the largest herbivores in Africa (*Ripple et al., 2015*). Although smaller mammals have a higher tick number to body mass ratio (*Hart et al., 1990*), potentially increasing the efficiency of tick harvesting, it seems that the absolute number of ticks and their abundance plays a more important role (*Mooring & Mundy, 1996*; *Koenig, 1997*; *Nunn et al., 2011*). Oxpeckers were identified as very effective tick removers from body parts that are inaccessible to self-grooming by host mammals, suggesting that they play an important role in tick control in the hosts (*Mooring et al., 2000*; *Bezuidenhout & Stutterheim, 2009*; *Nunn et al., 2011*; *Ndlovu & Combrink, 2015*). Furthermore, *Ndlovu & Combrink (2015)* suggested that oxpeckers may prefer larger-bodied ungulates because larger mammals can provide a more stable platform upon which to forage, or are just large enough to support simultaneous feeding of more oxpecker individuals. However, it is still possible that some variability in

this pattern may be explained by human preferences for large mammals, resulting in an overrepresentation of photos of large animals on online platforms (*Hausmann et al., 2017*). Our results also indicate that yellow-billed oxpecker visits a lower number of mammal hosts than red-billed oxpecker. This is in agreement with field studies showing that, even in localities where the distribution of both species overlaps (i.e., the spectrum of potential mammal hosts should be the same for both species), red-billed oxpecker has a wider range of hosts (*Ndlovu & Combrink, 2015*). However, we collected more photos of red-billed than yellow-billed oxpecker; thus, observed large-scale differences may be due to differences in sample sizes.

The nested structure of mutualistic networks is usually interpreted as an asymmetric specialization (i.e., specialists are associated with generalists rather than other specialists). The concept of nestedness has important consequences for ecological (*Bastolla et al., 2009*) and evolutionary (*Bascompte et al., 2003*) principles of biodiversity maintenance. In contrast to earlier results on a similar bird–mammal system in a Neotropical region (*Sazima et al., 2012*), our web structure for African birds and mammals from internet photos was only weakly nested. A relatively weak nested web structure, even for mutualistic relationships between oxpeckers and African mammals, compared with that between Neotropical birds and mammals (*Sazima et al., 2012*), could be explained by differences in diversity and spatial distribution patterns of large herbivores between both regions. In Neotropical region, only limited diversity of large herbivores is available, with few dominant and widely distributed species (e.g., domestic animals; see also *Sazima et al., 2012*); hence, bird communities visiting rarer species typically represent only a subset of bird communities visiting common species. In contrast, in the Afrotropical region, diversity of large herbivores is still relatively well preserved, with proportionally more widely distributed and abundant species than in Neotropical region. Therefore, it is possible that each mammal species is associated with own bird fauna, which is unique and species-specific, rather than being a subset of the bird species that can be found on some other mammal species, resulting in decreased level of observed web nestedness. However, it is noteworthy that commensalistic and mutualistic associations of birds with African mammals are even more common and include many more bird taxa than we reported (*Dean & MacDonald, 1981*). Our study was limited by the use of photos as information sources which enabled us to classify the exact activity of birds on mammal bodies, e.g., whether they are only sitting or also collecting some organic debris from the host. In mainly commensalistic birds, the level of nestedness was even lower and, moreover, nestedness did not differ significantly from a null model indicating that birds do not prefer mammal species which are visited by a large number of bird species. In oxpeckers, however, difference from a null model were found, suggesting that a nested structure may be the result of a non-random assignment of birds to their mammal hosts.

## Mammal and environmental correlates of bird visitation

We found that mammal body mass positively influenced the number and mass of birds observed sitting on mammals in the full set of species (i.e., taking oxpeckers together with other bird species). This may be simply explained by the fact that the size of larger

mammals can support a larger number of birds and/or birds with a higher mass which can drive a positive relationship between bird and mammal mass (*Kioko et al., 2016*). The latter explanation seems to be relevant, especially for the non-oxpecker set of species where mammal body mass was only significantly associated with the mass of birds. The majority of non-oxpecker birds associated with mammals were waterbirds, often heavy species when compared with, for instance, songbirds (*Maurer, 1998*). Finally, we found a positive correlation between mammal body mass and mass of oxpeckers. Oxpeckers live in small family groups (*Van Someren, 1951*) and large mammals may support the simultaneous feeding of larger family groups or other unfamiliar individuals on the same mammal individual.

Furthermore, our analysis revealed that larger mammal herd sizes were associated with a higher mass of birds and more bird species in the full set of species, and with bird mass in the set of non-oxpecker species. This may suggest that mammals in larger herds provide more feeding opportunities by flushing more potential prey, such as insects or small vertebrates, and, hence, attract a wider community of birds (*Wahungu, Mumia & Manoa, 2003*; *Kioko et al., 2016*). However, our finding of a relationship between mammal herd size and the number of associated bird species must be interpreted carefully. Despite weighting our regression analysis by the number of available photos per mammal species, such a relationship may be still mainly a function of a strong correlation between the number of bird species and sample size.

Habitat openness influenced the mass of birds and the number of bird species sitting on mammals; both were higher in more open habitats in the full set of species as well as in non-oxpecker species (see also *Kioko et al., 2016*). As mentioned above, the majority of diversity of bird species included in our study was represented by larger-bodied birds associated with water sources and relatively open country. We suggest that such patterns are most probably driven by habitat overlap between water- and open-country-associated larger-bodied birds and their mammal host because, similarly to birds, large African herbivores are more common in open areas than in closed habitats such as forests or woodlands (*Anderson et al., 2016*). Interestingly, the number of non-oxpecker birds was higher in more closed areas, probably because such habitats can provide more perching opportunities.

## Advantages and limitations of an internet search on the diversity of bird–mammal associations

We highlight the potential role of information technologies and new media, such as internet search engines, for further studies in ecology and evolution. Google Images represents one such resource that could facilitate a rapid collection of information on various aspects of ecological systems at large spatial and taxonomical scales. Results obtained from the analysis of internet images can be in good agreement with data from fieldwork, and such an approach could therefore, at least in some cases, supplement or replace less effective and time- and money-consuming fieldwork (e.g., *Leighton et al., 2016*; *Mikula et al., 2016*). While we revealed many previously described associations, our approach also identified many novel associations, e.g., that birds were also occasionally associated

with sitatunga *Tragelaphus spekii* and waterbuck *Kobus ellipsiprymnus*, and that red-billed oxpecker also feeds on much smaller hosts than previously reported (including Thomson's gazelle *Eudorcas thomsonii* weighing ∼20 kg) (for some details see Appendix S2; *Dean & MacDonald, 1981*; *Hart et al., 1990*; *Feare & Craig, 2010*; *Ndlovu & Combrink, 2015*). Moreover, similarly to *Dean & MacDonald (1981)*, we did not find any associated birds for several mammal species that we investigated including, e.g., beira *Dorcatragus megalotis*.

On the other hand, we were not able to find any evidence for several previously described associations, e.g., between chorister robin-chat *Cossypha dichroa* and nyala *Tragelaphus angasi* and bushbuck *Tragelaphus scriptus* (for other cases see also *Dean & MacDonald, 1981*). This may indicate limitations of our approach which may be the result of species- and region-specific public bias towards common or "charismatic" species of birds and mammals and to regions with a good infrastructure (*Clucas, McHugh & Caro, 2008*; *Hugo & Altwegg, 2017*; *Troudet et al., 2017*). Alternatively, the absence of some associations among the results of the internet search may be caused by our searching method because only English genus names were involved in the search, hence, overestimating associations for more common species. Finally, a sharp population decline or even extinction of large herbivorous mammals, mainly because of human-induced pressure, in many world regions, including Africa (*Ripple et al., 2015*), disrupts and reshapes present ecosystem services and ecological associations, including bird–mammal associations (*Galetti et al., 2017*; *Hempson, Archibald & Bond, 2017*). This may cause several bird–mammal associations to have been lost over time, resulting in decreased diversity of mutual associations. On the other hand, it is possible that other species which used to interact with herbivorous mammals have established new associations with mammals which are still common (*Galetti et al., 2017*). For instance, oxpeckers are able to behave plastically and can shift their host selection to other available herbivores, including domestic mammals in regions where they are common (*Dale, 1992*; *Feare & Craig, 2010*).

## CONCLUSIONS

We showed that using internet sources gives us an opportunity to access a large amount of data on bird–mammal commensalistic–mutualistic systems that has not been previously studied with such a wide perspective. We found that mutualistic webs which included only oxpeckers were more nested than the mainly commensalistic webs which included other species. Moreover, we found support for the idea that patterns of associations between birds and large African herbivorous mammals can be linked to mammal and environmental characteristics, including mammal body mass and herd size, and habitat openness. Further studies could focus more on birds that associate with mammals without regularly sitting on them; it is probable that the restriction of our analysis only to photos showing birds sitting on mammals eliminates many commensalistic and sometimes also mutualistic associations of birds with mammals. Sitting or feeding associations between African birds and herbivorous mammals in savannahs are only one aspect of commensalistic–mutualistic associations between birds and mammals. Similar interspecific relationships have also convergently evolved between birds and, for instance, larger-bodied terrestrial mammals

in Neotropical regions (*Sazima et al., 2012*), monkeys (*Heymann & Hsia, 2015*), otters (*D'Angelo & Sazima, 2014*), dolphins (*Bräger, 1998*), and domestic animals such as cattle which replaced native herbivorous mammals in many world regions (*Kioko et al., 2016*; *Galetti et al., 2017*). Our results could thus also bring new insights into the complexity of bird–mammal associations in other world regions and systems including different animal taxa. However, further comparison with *in situ* observations of commensalistic–mutualistic systems to check for potential biases is strongly required.

## ACKNOWLEDGEMENTS

We are very thankful to Martin Hromada for stimulating the debate over this topic and Tim Sparks for English language correction. We are also thankful to Federico Morelli for help with data visualization.

### Funding
The study was financially supported by the Czech Science Foundation (14-36098G). The funders had no role in study design, data collection and analysis, decision to publish, or preparation of the manuscript.

### Grant Disclosures
The following grant information was disclosed by the authors:
Czech Science Foundation: 14-36098G.

### Competing Interests
Piotr Tryjanowski is an Academic Editor for PeerJ.

### Author Contributions
- Peter Mikula conceived and designed the experiments, performed the experiments, analyzed the data, contributed reagents/materials/analysis tools, prepared figures and/or tables, authored or reviewed drafts of the paper, approved the final draft.
- Jiří Hadrava analyzed the data, contributed reagents/materials/analysis tools, prepared figures and/or tables, authored or reviewed drafts of the paper, approved the final draft.
- Tomáš Albrecht and Piotr Tryjanowski conceived and designed the experiments, authored or reviewed drafts of the paper, approved the final draft.

### Data Availability
Data is available in the Supplemental Information.

### Supplemental Information
Supplemental information for this article can be found online at http://dx.doi.org/10.7717/peerj.4520#supplemental-information.

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
