# Peer review of "Large-scale assessment of commensalistic–mutualistic associations between African birds and herbivorous mammals using internet photos"

_PeerJ, doi:10.7717/peerj.4520_

## Round 0.1 · original submission · Major Revisions

· Academic Editor

Major Revisions

Both reviewers found your article original and potentially publishable but needing substantial revision. They provided numerous helpful and insightful suggestions to improve the manuscript. I also had quite some concerns, some not mentioned by the reviewers. Because I do not have direct experience with this system, it is possible that I did not recognize some assumptions or concepts that the reviewers shared with you. However, I think that my experience level might be similar to that of many reviewers, so that my failure to understand some issues may indicate that they need more elaborate explanation for a broader audience. In addition, I noted many minor errors in style, grammar, punctuation and word choice.

Below, I provide my suggestions as Editor's Comments. I apologize for the number of comments but hope you find them useful. You may treat these as a third review. That is, make appropriate changes if you think my comments are valid. Otherwise, provide a detailed explanation of your reasons for not making changes.

For the minor grammatical and stylistic issues, I have provide an annotated pdf with yellow highlights to indicate problems and inserted comments to either explain the issue or provide an alternative text. Some of the highlights do not have explanations because I thought they were obvious, had been noted earlier in the manuscript or were included in my written comments below. For the manuscript annotations, you do not need to provide a complete response for each point; however, if you disagree and are not making changes, please indicate the reasons in your letter.

Finally, I also felt the need to elaborate on three of the reviewers' comments. These 'Notes on Reviewers' Comments' are listed after my 'Editor's Comments'.

Editor's Comments

General concerns
Introduction
• The first paragraph that focuses on the savannah seems unnecessary and even misleading. The research objectives seem much more tied to relationships between birds and mammals, and non-savannah habitats (forest) are included in the study. I feel that the logical development of the manuscript would be stronger if it started with a set of expected patterns expanded from the current paragraphs 2 and 3 (L39-69).
• Parallel to the development of potential relationships between body size, herd size, and environment, the Introduction needs to introduce the concept of interaction webs and nestedness, what they are and what questions arise from their potential application to this system.
• What you mean by the term 'commensalistic-mutualistic' needs to be clearly defined and linked to the study system. A brief definition of each concept would be helpful, along with how you want the reader to interpret the hyphenated term. As you describe the system and behavior, more attention to the possible or established benefits and costs of each partner would be helpful. While you discussed the oxpeckers in some detail, you did not provide equivalent attention to birds that perch on mammals.
• I agree with the reviewer that the paragraph in L70-81 is not clearly linked to the objectives, despite the last line. The paragraph should be removed or the logical relevance made much clearer.
• The objectives need to be much more explicitly defined, with explicit statements of dependent and independent variables and the goals of the interaction web component of the study. Referring to 'patterns' is too vague.
• To aid reader comprehension, organize the objectives such that the sequence of objectives matches the order of methods, results and discussion.
Island Biogeography
• It is not obvious to me that considering herds of mammals as islands (L43, 184, 406ff) is a useful or even valid concept. Many of these herds are based on fission-fusion social systems that are unstable in terms of composition and both relative and absolute location, and, in contrast to island biogeography, the birds don't immigrate to and establish populations on these herds or individuals (as far as I am aware). I think that a framework of resource patches at two scales (individuals and herds) would be more appropriate from a foraging perspective. How this perspective would apply to use of mammals as perches is less clear.
Optimal foraging
• L61-62. Charnov's model is about patch-leaving rules and its application requires much more information than what you provide here. In the general optimal foraging literature, you could find more appropriate references, if needed, for the concept of resource patches. Why not simply state that the distribution could follow tick abundance rather than body size or herd size per se? At present, the concept of optimal foraging is presented in the Introduction and briefly mentioned in Conclusions, but there is no robust argument for its relevance. Either develop the optimal foraging interpretation of the data more fully in the Discussion or remove it from the Abstract L16.
Interactions?
• In the Title, Abstract, Introduction, and many places elsewhere, you refer to this as a study of 'interactions'. However, it seems to me that you do not investigate interactions as usually defined (reciprocal actions or influence). I wonder if another more neutral terms, perhaps 'association', would be more appropriate.
Species descriptions
• L328 and elsewhere. I think that the manuscript would be easier for most readers if you used English names for all species, including the oxpeckers, providing scientific names at first use.
Sampling methods
• L117. Part of the search procedure is unclear. The reference to 'each species of mammal' implies that you incorporated all species in an individual search rather than searching for the birds associated with a particular species. However, 'bird taxon' implies that you only searched for one bird species for a given taxon. These implications are contradicted by L122-124. Reorganize L117-126 to reduce redundancy and increase the clarity.
• L121. It seems to be that a search for only the most probable taxon followed by more general terms such as 'bird' only if nothing was found is likely to bias the search in favor of common or well-recognized associations.
• L122-126. The search effort and its implications needs some clarification. First, if you searched each combination until you found few new photos, it seems to me that the number of photos in your data set would be proportional to the number on the internet. The number of photos on the internet would seem likely to be influenced by the range and abundance of the bird and mammal species involved, accessibility and photographic conditions in the areas where they occur, and some measure of charisma that attracts tourists or researchers. Thus, I cannot understand your statement that this removed biases due to abundance and distribution (L129). Furthermore, I don't think equal effort is the correct term for this procedure because species with a lot of photos will take more effort (photos evaluated and saved) to reach the stopping point than species with few photos. Then, the statement L126 that you did not stop at the equivalent point for the most common species seems to contradict your procedure. It seems to me this part of the methods needs to start with a statement of what the objective of the procedure was (for example, a record of the relative frequency of each association), followed by a description of the procedure, and then a clear indication of what situations resulted in deviations from the procedure. The Discussion should include evaluation of how this procedure and underlying influences on photo frequency could influence the observed patterns.
• L198-202. I think the reader needs more information about how you distinguished the habitat structure categories 2, 3 and 4. Are the terms in parentheses actual technical terms for vegetation categories? If so, please provide a reference so that readers can determine their characteristics. If they are not technical terms (and perhaps even if they are), provide the operational definitions you used so your study would be repeatable by another researcher. For example, based on my experience in Africa, I might have characterized savannah as open habitat.
• L203-205. It is not clear why you define categories of habitat structure and then covert them into openness in the same series. Why not simply state that you defined surrounding habitat into four levels of openness and then describe what vegetation types were included in each category? I assume that you used only integer values and not fractions, but that should be stated explicitly.
Data analysis
• After the first two readings, I was not confident that I understood your dependent variables in the PGLS analysis. Your most explicit statement (L290-292) does not explain precisely what these measures are (i.e., the units and how they were calculated from your data). Going back to the prior information about the variables (L171-177) did not help. Indeed, bird mass is not even mentioned there. You state that your unit is an individual mammal and the birds on it (L171). This suggested that each of these units was a point in your data analysis. However, bird species richness was defined at the mammal species rather than individual mammal scale. After multiple readings and checking Table 1 and Appendix S5, I am pretty sure that your analysis was at the scale of mammal species with the individuals being used to calculate species means, but I feel that you need to make these variables much clearer. If my understanding is correct, your variables might be clearer as 'mean number of birds per individual', 'mean total mass of birds per individual' and 'total number of bird species per mammal species'. Once these are clearly defined, referring to number of birds, mass of birds, and number of species of birds would be acceptable in the text and tables.
• Species richness measures typically shows a sample size effect. As sample size increases, richness increases before reaching an asymptote. Therefore, to compare richness in different contexts, a rarefaction analysis is required to ascertain that any difference is not due to sample size. You may be able to do a rarefaction analysis for some of the most common species, but your samples will be too small for others. Therefore, I think at least you need to look at whether species richness is correlated with sample size and consider the implications of that interpretation in the Discussion. This confound may also affect your discussion of oxpeckers since the supposedly more generalist species has a much larger sample size. Reference: N. Gotelli and R. Colwell 2001 Ecol Lett 4:379-391 and many citations to this article.
• How much of the variation in your data set was explained by your predictive models? In some places, you imply a strong causal effect, e.g., 'bird-mammal interactions may be determined by simple predictions' (Abstract). The variance explained, strength of the correlations, and potentially confounding variables (such as perhaps a correlation between mammal abundance and herd size) need to be considered in the discussion before such strong statements can considered.
• I think you need to be explicit that you did not include individuals without birds (zero values) in your data set, even if such individuals were visible in photos that did include birds on some animals.
Frequency
• You put substantial effort into assuring that your samples were equivalent among mammal species. However, I don't think that you make as much use of these data as you might. Although they appear in the network figure, you don't make much use of them other than the mention of oxpeckers being associated with larger mammal species. Isn't there anything else you can say from these data?
• With regard to the association between oxpeckers and larger mammals, you need some statistical support to show that this is not a random association.
Other results
• The Discussion (L348) refers to the discovery of some novel associations and failure to detect some recognized ones. This material needs to be stated in the Results, with the specific associations stated.
• Were there any species of mammal that would have been included in your study but did not have any associated birds?
Figures and tables
• It seems to me that the summarized data resulting from your search (Appendix S5) would be appropriate for a Table in the main text. It would be great to add a code so that the bird species could also be included. Number of photos could easily be incorporated too. This would give readers a much better overview of your data set. Whether a table in the article or appendix, there need to be units in the column heads. There also appear to be unrealistic levels of precision in mass and other variables in the Appendix table.
• Check whether the issues of clear column heads, units and levels of precision may apply to other supplementary tables also.
• Some minor changes are proposed for Table 1 and Figure 1.
• Figure 2. Considering that it can be a hassle to rotate a page when reading on a computer, I think it would be clearer to rotate the figure such that the species names are horizontal. You might also switch to the English names of the mammal and bird species if you agree with my earlier suggestion. Are the bars truly the frequency as stated in the caption or do they represent a proportion of the total? If the latter, what is the total? In either case, I think a key to indicate frequency or total is needed. You also need to define the gray lines joining mammals and bird species.
• A new figure? Even though your analysis is multivariate, it is possible that a graph showing univariate relationships between some of the main independent and dependent variables (including but not necessarily limited to the ones with particularly strong patterns) might be of interest to readers.
Discussion needs substantial revision
• While a brief overview of your Results may help the reader focus on the issues for discussion, I think the first paragraph of the Discussion is too detailed. Furthermore, rather than strictly summarizing results, it makes statements about inferred conclusions that require additional interpretation of the results. In particular, the statement that the most important factors affecting mammal-bird interactions were body size and herd size of the mammals. This statement requires a discussion of effect size rather than simply p-values and turns a correlational value into a causal one, needing a specific argument and addressing potential confounding effects such as correlations between measured variables and other possible causal variables and limitations or biases in the data.
• I suggest that the Discussion include a paragraph for each of your independent variables, considering reliability of your conclusion and potential causal processes involved, both for the variables that were statistically significant and for those that were not. Because your three dependent variables are related, a coherent discussion of why some and not others might be affected would be easier if you grouped by independent variables. Issues such as discrepancies between the complete data set and the one that included only species with 10 or more photos would be much better addressed in these separate sections. For example, you could see which species drop out in the limited data set and whether this loss might help explain the change in strength of pattern. The brief review of the results would come at the start of each of these paragraphs rather than combined at the start of the Discussion. At present, this is done briefly for habitat openness but the third paragraph of the Discussion then goes in other directions.
• As noted above, I strongly disagree with the application of island biogeography to this system because the birds are likely to be only short-term residents on individual mammals. The fact that the concept has been applied to other size scales is not sufficient to justify this extension because these appear to be more likely to involve immigration and population change. If you wish to include this concept, you need a much more rigorous discussion.
• I found your discussion of the association of oxpeckers with larger mammal species but lack of an effect of mammal size on number or mass to be interesting. However, your Results did not provide statistical support for the first relationship, and your Discussion did not address the nuances of this effect such as that your data could not include mammal species that were never associated with birds (limiting your measure of species with which oxpeckers did not associate).
• With regard to the application of optimal foraging to this pattern, I do not think you have made a strong enough inference for it to be mentioned in the Abstract. It is possible that with a more developed analysis of the observed distribution and a more sophisticated application of foraging theory such a point might be possible. For example, I would have expected foraging rate to be associated with ticks per unit surface area rather than body mass and that the cost of finding the next host and moving to that host should be considered. Furthermore, it is well recognized in foraging theory that animals are not always found in the richest patches because competition results in a broad distribution (e.g., the Ideal Free Distribution). The brief comment in the Conclusion should be part of this discussion rather than separated from it.
• Why is there no discussion of the interaction web data? If it is worth doing the analysis and presenting it, it must be worth discussing.
• The paragraph on nestedness is rather confusing. This may be because the reader has not been offered a clear understanding of the concept in the Introduction. For example, how is low nestedness in Africa explained by low mammal species richness in South America?
• Because the application of your study to conservation and ecology of savannahs is quite indirect, I suggest consolidating these comments into a short paragraph late in the Discussion or restricting it to a couple of sentences in the Conclusions.

Other suggestions and comments
• Is the term 'megafauna' useful in this context? It seems to me that you are including some species much smaller than what others consider as megafauna and you are not including carnivores that might be considered megafauna. Why not just refer to 'African birds and mammals' in the title?
• What about the term 'host'? Is there a well-established use of this term for the oxpeckers (as opposed to the ticks) and has it been applied to perching as well? I am not against use of the term, but you need to inform the reader if you are using an established word in a new context.
• It is a good idea to include the Abstract after the title page in the manuscript itself so that it has line numbers for easy reference.
• Abstract L3. I don't think there is a difference between sitting and perching. Delete one of these unless you make the distinction clear later in the manuscript.
• Abstract L7. Make the concepts and relevance of interaction web structure and nestedness clear.
• Abstract L12. It is confusing to change the order of dependent and independent variables in successive sentences.
• Abstract L13. Give common as well as scientific name of oxpeckers.
• Abstract L13-15. Comparison incomplete and terminology unclear.
• L46. Use another expression such as 'load-carrying capability' because 'carrying capacity' is a concept in population biology different from what you intend here.
• L57 'exclusively obligate' appears to be redundant.
• L105. You did not really investigate the character of interactions.
• L116. There is no need to mention particular countries. Many African countries are French-speaking and this might be a larger restriction than the two Portugese-speaking ones. Besides, I would imagine that people who visit English-speaking countries will not necessarily post in English or use the English names.
• L179 and elsewhere. Use 'mass' rather than a mix of mass and weight.
• L184-188. This information belongs in the paragraph starting on L134.
• L191. This sentence relates to your criteria for adding a photo to your data set and also belongs in the paragraph starting on L134.
• L209ff. This section is not clear because the reader does not understand the objective of the analysis. The background information should have been presented in the Introduction and there should be a specific objective (or more, if nestedness is separate) related to this topic. Here, all you would need to do is briefly refer back to the objective as you introduce the method.
• L217ff. A separate paragraph for nestedness seems appropriate. The background material except what is specifically related to methods belongs in the Introduction.
• L223. You have not mentioned separate analyses for the oxpeckers.
• L229ff. This paragraph explains why you did not have a preference objective. It needs only one sentence and belongs in the part of the Introduction where you develop the rationale for your objectives.
• L238ff. The section on zoom includes information about scoring the photos. This information belongs with the other information about scoring such as number of birds, herd size, habitat, etc.
• L245ff. This section discussing lack of bias in herd size and bird number and total mass in relation to zoom level is appropriate for the analysis section. However, you should present it clearly as a series of analyses to test the effect of zoom followed by a statement that you did not find an effect.
• L260ff. This paragraph discussing lack of effect of zoom on 'observed patterns' is too vague to be understood. Judging by Appendix S3, these are the patterns for which you used the PGLS which has not yet been presented. What are the independent and dependent variables? Wouldn't it make more sense to present this after the PGLS to show that there was little or no effect of zoom? Why does this analysis not include the phylogenetic aspect?
• L276. Reference needed here, even though you do cite references later in the paragraph.
• L287. I don't understand how a single index can indicate phylogenetic dependence at high value and lack of phylogenetic relatedness at low value. These sound like different concepts, although I am not sure I know what lack of phylogenetic relatedness means.
• L290. If you ran models separately for each of the three bird characteristics, as I assume you did, writing 'the variables were' would be clearer.
• L291. Be sure to keep the variables in the same order consistently in methods and results, including tables.
• L293. I don't understand how the similarity of weight in oxpeckers results in exclusion of number and species richness as dependent variables. Overall, it seems that these could have been included.
• L300. This does not seem to be a prediction that you are capable of testing. I think you might mean 'we assume'.
• L332. This statement is not correct. It implies that there is a set of birds associated with water bodies in general and that nearly all were found on hippopotamuses. I think you mean something like 'Nearly all bird species in the data set that are associated with aquatic habitats were photographed perched on hippopotamus'. However, you need to provide more support for this, such as identifying the birds associated with water because the reader otherwise needs to know the habitat associations of all birds in the data set.
• L339. Again, the reader cannot assess this statement without information on which are the very small species.
• L343. Did your statistical methods section mention that you calculated nestedness separately for non-oxpecker and oxpecker data sets and explain why this grouping was different from the one you used for the phylogenetic regression?
• L360. The relationship with openness was negative.
• L364. This sentence is confusing. Why are there two log-likelihood values for what seems to be one relationship? The table and methods seem to imply that you only looked at total bird mass (although I don't see why you excluded number), but here you refer to richness.

Editor's Notes on Some Reviewers' Comments
• Reviewer 1 suggested avoiding 'e.g.' and 'see also' with citations. I don't quite agree with this view. To me, e.g. implies that the author is providing one or more examples, not necessarily the first or best. A reference without 'e.g.' is meant to be a more definitive source, such as the first, clearest or most recent presentation of a method or concept.
• Reviewer 1 suggested avoiding stating that your study is the first to do something, implying that it may lead to challenges for readers or other authors. In contrast, I feel that is important for authors to identify the originality of their contribution. Perhaps, qualify the statement with 'to the best of our knowledge'.
• Reviewer 2 suggests distinguishing mutualistic and commensalistic relationships more clearly. In referring to 'mixing up' these concepts, I believe that he does not mean that you had the meanings wrong (mixed up) but that you combined them (mixed). In part, this comment is related to mine on clarifying the concepts. I believe that you have some statements indicating the difficulty of separating species by type of relationship. If you cannot find a way to address the distinction between these relationships in your analysis, you should at least more explicitly explain and justify combining them and consider any implications for the interpretation of your findings.

·

Basic reporting

There are a few places where sentences didn't make sense and minor rewording is needed, but in general, the paper was clear, well written, and structured fine.

Experimental design

The authors use a novel source of data to examine relationships between African birds and mammals. In general, the aims of the study were clear and the methods adequately described.

Validity of the findings

Mostly fine, although I have a couple of suggestions.

Additional comments

This is an interesting use of citizen science data to investigate the apparent mutualistic relationships between birds and mammals in sub-Saharan Africa. Their use of data has various limitations, most of which the authors seem aware of and deal with appropriately. The one variable I would recommend they delete from the analyses is elevation, unless they can generate some a priori hypothesis for why it might be important. I also still don’t understand what “nestedness” means, exactly, in the context of their analyses. Since this plays a fairly important role in their discussion, I would recommend they try to explain it better for those of us who are not familiar with this concept. In general, however, I think the authors did an interesting job of using photographs to infer and in some cases even test prior relationships between African birds and mammals.

Specific comments:

Lines 28-32. I don’t think it’s necessary to devote space telling readers that the savanna is not uniform.

Lines 70-81. Most of this seems peripheral (at best) to the paper. Delete or reduce.

Line 76. Not quite a sentence…reword (“This may cause that…”)

Lines 91, 98, 168. Avoid “e.g.”s and “see also”s; all citations are but representative of those that could be cited.

Line 99. Avoid “the first”; maybe it is, maybe it isn’t, but it’s asking for it in a way that’s unnecessary.

Lines197-198. The suggests that when an exact location was not specified, the elevation of whatever the center of the species in that country was used. This seems way too likely to introduce potential bias. I would exclude such cases when it comes to elevation and habitat. Actually, it’s unclear to me why elevation was analyzed at all; what is the rationale for including it as an explanatory variable? Given that it’s not significant anyway, I’d suggest throwing it out.

Lines 208-228. I understand the webs, but I don’t understand how this particular one could be nested, or what, exactly, a “nested web” means or would look like in this case. Please clarify.

Lines 248, 257. Round r value to 2 decimal places.

Line 253. “avoid” or “circumvent” rather than “eradicate”.

Lines 342-345; 353-357. Round off these and other values to 2 decimal places, please.

Lines 341-347. Again, I don’t really understand what “nestedness” means in this example.

Line 352. Please clarify what “a negative relationship with habitat openness” means.

Lines 374-375. Unclear what “higher bird mass” refers to, exactly.

Lines 411-412. Reword (“In result…”); also delete second MacArthur and Wilson references.

Lines 447-457. This helps but it’s still not clear to me what nestedness would look like in this situation.

Lines 475-480. This strikes me as far less likely than that sampling by photographers is incomplete (the author’s alternative explanation). I’d delete. It's speculation at best, but speculation that is too important to throw out without better evidence.

Reviewer 2 ·

Basic reporting

1. Language
The English is generally fine, but could still be improved. At a number of places, the article is missing. Throughout the text, weight should be replaced by body mass (currently there is a mix of both terms).

2. Background/context and references
While the manuscript makes a few cross-references to work performed outside Africa, it ignores important recent literatur.
The recent book by Goodale, Beauchamp & Ruxton (2017; Mixed-species animal groups: behavior, community ecology and conservation) provides a broad and comprehensive background to interspecific mutualistic and commensalistic interacts. Consulting this work will help to draft a more focussed background and sharpen the research question (see below).
Heymann & Hsia (2015; Biol Rev) reviewed the interactions of primates and other animals. Although they did not perform sophisticated statistical analyses, they also examined the influence of various factors (e.g., body size, group size) and reached some conclusions that might be relevant for the discussion.

3. Article structure etc.
The manuscript meets the expectations with regard to the structure and figures. Raw data are provided as supplements.
Table 1 needs to be improved. The two separate sets of results (for full data set and for restricted data set) are not visually clearly separated. It took me a while to match the information in the text with the content of the table. I suggest to use additional headers in the table, one over the columns referring to the full data set, and another over the columns referring to the restricted data set

4. Self-contained ...
This is a major weakness of the manuscript. Hypotheses and predictions are not explicitly stated. This strongly diminishes the value of the manuscript, as it makes it more difficult to understand the results.

Experimental design

1. Aims and Scopes
The manuscript fits with the aims and scopes of the journal.

2. Research question ...
Examining patterns of interspecific mutualistic and commensalistic interaction is a rapidly expanding field in ecology, with broad implications for understanding e.g. ecosystem functioning, adaptations and conservation. With regard to interactions between mammals and birds, there is a large amount of mainly anecdotal information, but very few reviews and quantitative analyses. Therefore, the topic of the manuscript is of considerable interest and could potentially contribute to filling gaps in our knowledge.
However, in its current form the manuscript shows a number of problems, identified in the following:
a. Mutualism and commensalism are mixed up. Certainly, there is no strict limit (as is most often the case in biology) between categories created by scientists. For some interactions between organisms, e.g. ant-plant interactions, the interaction type may even range from mutualism to parasitism. Nevertheless, it would be useful to distinguish mutualistic and commensalistic interactions and to categorize the species combinations accordingly. Obviously, the "cleaning" interaction between oxpeckers and mammals is clearly different (both parties generally benefit) from the "sitting" interaction between e.g. egrets and mammals (only the birds benefit). Mixing up mutualistic and commensalistic interactions is as problematic as if one would mix-up seed dispersal and pollination in animal-plant network analyses.
b. As I understood the manuscript, analyses are restricted to those interactions where the birds sit (either for sitting or for cleaning) on the mammals, but excludes birds flying around mammals. Altough the criterion to restrict the analyses to photos showing birds sitting on mammals has the advantage of using a simple criterion, it probably eliminates many (commensalistic) interactions of birds that associate with mammals without regularly sitting on them. I think that this needs to be more strongly considered in the conclusions.
c. The research question is not very well defined. It took me two readings of the Introduction to (hopefully) understand what the authors actually intended to study. This problem is aggravated by the fact that the authors unnecessarily introduce reasons for their analyses that are honourable - namely conservation of savannahs and of the interactions therein -, but that are scientifically not really needed to justify the study. Furthermore, the justification in terms of conservation is partially not correct: when the authors state (l. 71) that "Savannahs are one of the most threatened world", this is only correct for temperate but not for tropical savannahs (Conservation Risk Index 10.1 for the former, 2.0 for the latter; see Fig. 1 in Hoekstra et al).

3. Rigorous investigation ... & Methods ...
a. As mentioned above, the lack of distinction between mutualism and commensalism is a basic problem.
b. Otherwise, the procedures for selecting the photographs from the internet are well described and allows for potential replication.
c. While the authors include geographic information, they do not correct for the potential effect of spatial autocorrelation. Many photos are likely to come from the same sites frequently visited by tourists and scientists. This may inflate numbers for certain species combinations. The lack of controlling for this potential bias is in marked contrast to the otherwise sophisticated statistical analyses, including phylogenetic corrections.
d. It did no become clear to me what your sample unit is. As I understood, it is one interaction, i.e. one or more birds sitting on a mammal. But it is not clear whether photos showing several mammals, each with one or several birds from the same species, are scored as one or as several interactions (the latter violating statistical independency).

Validity of the findings

Given the problems identified above, it is difficult to evaluate the validity of the findings.

As with the justification of the study, there is also some overstament with regard to conservation in the Discussion. As long as it is not known which proportion of resources birds derive from the interaction with mammals (except for the obligatorily interacting oxpeckers), the conservation implications of the drop-out of interaction parners cannot be evaluated, in contrast to e.g. seed-dispersal mutualism where the effects of drop-out have already been robustly demonstrated.

Additional comments

Despite my criticism stated above, I like the approach taken by the authors. Commensalistic interactions may play larger roles in ecology than is generally appreciated, but we are far from understanding their impact. Therefore, analyses that tackle patterns of commensalistic interactions are highly welcome. I think there is a large potential for making an important contribution to the understanding of interspecific interactions between birds and mammals once the problems addressed above have been solved.

---

## Round 0.2 · accepted · Accept

· Academic Editor

Accept

The manuscript has been greatly improved and is now ready for publication with the exception of some small corrections of English word use, punctuation and sentence structure and the replacement of Fig 2 and Appendix S3. I have provided a pdf with highlights to indicate words that need to be changed and inserted comments to suggest alternatives.